# Uncommon and Rare *EGFR* Mutations in Non-Small Cell Lung Cancer Patients with a Focus on Exon 20 Insertions and the Phase 3 PAPILLON Trial: The State of the Art

**DOI:** 10.3390/cancers16071331

**Published:** 2024-03-29

**Authors:** Federico Pio Fabrizio, Ilaria Attili, Filippo de Marinis

**Affiliations:** 1Laboratory of Oncology, Fondazione IRCCS Ospedale Casa Sollievo della Sofferenza, 71013 San Giovanni Rotondo, Italy; federico_fabrizio@hotmail.it; 2Department of Experimental Oncology, IEO European Institute of Oncology IRCCS, 20139 Milan, Italy; 3Department of Oncology and Hemato-Oncology, University of Milan, 20122 Milan, Italy; 4Division of Thoracic Oncology, European Institute of Oncology, IRCCS, 20141 Milan, Italy; ilaria.attili@ieo.it

**Keywords:** EGFR, NSCLC, amivantamab

## Abstract

**Simple Summary:**

The dramatic improvement in the prognosis of patients with advanced epidermal growth factor receptor (*EGFR*)-mutant non-small cell lung cancer (NSCLC) became possible thanks to the advent of EGFR-tyrosine kinase inhibitors (EGFR-TKIs). In a subgroup of *EGFR* mutations, known as uncommon (ucEGFRmuts) and rare, NSCLC-mutated patients show, most of the time, lower EGFR-TKIs sensitivity than most common mutations, making this a great clinical point of discussion. Here, we summarized recent data about *EGFR* exon 20 insertion-positive NSCLC patients and Phase 3 trials ongoing, with a specific focus on the PAPILLON study.

**Abstract:**

Uncommon (ucEGFRmuts) and rare epidermal growth factor receptor (*EGFR*) mutations account for 10–15% of diagnosed cases and consist of a heterogeneous group represented by several clusters within exons 18–21 (e.g., exon 18 point mutations, exon 21 L861X, exon 20 S768I), as well as exon 20 insertions (Ex20ins). Their incidence is under molecular and clinical investigation following recent findings that reported an increase of sensitivity and specificity of next-generation sequencing (NGS) methods. Consequently, their detection allows for the selection of emerging treatment options to significantly improve patients’ outcomes in these particular subgroups of *EGFR*-mutated advanced non-small cell lung cancer (NSCLC). Specifically, this commentary is focused on the notable progress of the Phase 3 PAPILLON study that showed primary efficacy results from amivantamab, a bispecific antibody with specific binding and affinity to extracellular domains of EGFR and MET, plus chemotherapy in the first-line setting for EGFR exon 20 insertion–mutated advanced or metastatic NSCLC patients, as compared with chemotherapy alone, thus becoming the new standard of care in this group of patients.

## 1. Introduction

Epidermal growth factor receptor (*EGFR*) is a transmembrane protein that influences the pathogenesis of non-small cell lung cancer (NSCLC) by activating cellular signaling networks, such as the Ras/Raf/Mitogen-activated protein kinase/ERK kinase (MEK)/extracellular-signal-regulated kinase (ERK), PI3K/PTEN/Akt/mTOR (phosphatidylinositol 3-kinase/protein kinase B/mammalian target of rapamycin), and Jak/STAT (Janus kinase/signal transducers and activators of transcription) pathways, leading to tumor cell proliferation, invasion, and metastasis [1]. The most recurrent and activating *EGFR* mutations, also defined as classical mutations, are small in-frame deletions within exon 19 (E19del) and L858R exon 21 of somatic origin that results in a leucine to arginine amino acid change at position 858, which causes constitutive activation of the EGF receptor [2]. Actually, they represent about 80–85% of the total cases of *EGFR* mutation-positive lung adenocarcinoma (ADC) patients (patients with mutations located in the tyrosine kinase domain of the aforementioned gene) and they have a high sensitivity to EGFR-TKIs [3,4]. On the other hand, uncommon (ucEGFRmuts) and rare *EGFR* activating mutations account for 10–15% of diagnosed cases in Caucasians and about 45–50% of Asian populations, and they consist of a heterogeneous group represented by several clusters within exons 18–21 (e.g., G719X including G719S, G719A, G719C, and G719D substitutions; S768I; and L861Q in exons 18, 20, and 21, respectively), as well as exon 20 insertions (Ex20ins) with a typical recurrence in adenocarcinoma histology and nonsmoking women [5,6,7].

Recent breakthroughs with the advent of next-generation sequencing (NGS) applications have markedly improved the sensitivity and specificity of the detection of oncogenic drivers in the complex landscape of patients with NSCLC [8]. The understanding of molecular and genomic features has changed the way of diagnosing and treating this heterogeneous disease, allowing for the selection of more effective and personalized treatments in several subsets of *EGFR*-mutated NSCLC patients [9]. This commentary reports the real possibility of detecting and treating ucEGFRmuts that, until recently, represented an unmet need, to the best of our knowledge and on the basis of the papers that we deemed important to discuss.

Despite being less representative, these mutations are associated with a lower sensitivity to EGFR-TKIs, with most data being available on the clinical activity of afatinib against ucEGFRmuts [5,10,11].

Moreover, the effectiveness and safety of afatinib was determined by Yang JC and colleagues in a combined post hoc analysis of a single group phase 2 (LUX-Lung 2) trial and a randomized phase 3 (LUX-Lung 3 and LUX-Lung 6) clinical trial with *EGFR* mutation-positive advanced (stage IIIb–IV) lung adenocarcinomas patients, reporting fascinating results. Particularly, afatinib demonstrated clinical inhibitory activity in terms of an extended median PFS (progression-free survival), OS, and ORR (objective response rate) against the major ucEGFRmuts and rare *EGFR* mutations, among which G719X, S768I, and L861Q, when compared to the other ucEGFRmuts for the aforementioned gene, were more prevalent [12].

The clinical activity of afatinib versus the most prevalent ucEGFRmuts has been consistently reported in several retrospective studies, thanks to which it was possible to obtain an ORR of around 70% in a subgroup of compound/complex mutations, with a substantial decrease of about 20% for cases of exon 20 insertions [10,13].

Focusing on the molecular aspects of *EGFR* rare and ucEGFRmuts, an interesting systematic literature review about their prevalence and clinical outcomes among locally advanced/metastatic NSCLC patients was recently reported by John T and colleagues. Based on ten studies, they identified an occurrence rate of these mutations in a variable and large range, between 1% and 18%, as follows (from lowest to highest percentage for all *EGFR* investigated mutations): S768I in exon 20 (0.5–2.5%), L861X in exon 21 (0.5–3.5%), Ex20ins (0.8–4.2%), and G719X in exon 18 (0.9–4.8%) [5].

In the spectrum of the genomic *EGFR* alterations, from a total of 237 tumor samples tested by NGS analysis, Mehta et al. identified sixty-nine (~29%) *EGFR* mutated cases, of which forty-one (~59%) had the most recurrent activating *EGFR* mutations (approximately 22% for p.L858R and 38% for Del19). Moving on to the ucEGFRmuts and rare aberrations, it would certainly be worth mentioning the occurrence of *EGFR* amplification in six patients (8.7%), two of which harbored *MET* exon 14 skipping. Moreover, exon 20 insertions (7.2%) were present in five patients. Interestingly, in a small fraction of less than 3%, two patients that had exon 18 indels (i.e., pE709_T710delinsD, 2.9%) and were treated with afatinib demonstrated a good survival response, in line with previous preclinical research [14,15].

A large cohort of 5363 Chinese lung cancer patients was subjected to genotyping for the detection of *EGFR* mutations, in which the frequency of the common/typical mutations appeared to be about one-third (34%) compared to ucEGFRmuts (12%). In this latter subgroup, retrospective clinico-pathological data showed a better EGFR-TKI response, being particularly good in patients with G719X and compound L858R mutations, suggesting its importance as a first-line therapy [16].

Recently, Bar J and colleagues reported, in a total of 60 patients with a predominantly Caucasian population, the largest screening of *EGFR* mutational subgroups, through which it was possible to identify previously unreported Thr790Met (T790M, 15%), L861Q (20%), and G719X (30%) mutations, although these latter two are known to confer sensitivity to EGFR-TKIs, but with a less effective response compared to typical *EGFR* mutations [17,18]. These real-world molecular data provide a strong clinical relevance, in particular for the use of first-line osimertinib, in a subset of patients with ucEGFRmuts, since this third-generation EGFR-TKI demonstrated a high rate of disease control [18].

To date, the detection of ucEGFRmuts is becoming less complex because of the evolution of NGS technologies that allowed its introduction into clinical practice. The prevalence of this subgroup of *EGFR* mutations, differently distributed worldwide [19], could significantly vary in relation to the efficacy and sensitivity of profiling tests and type of reports, considering their potential roles as single drivers or presence within compound mutations [20,21].

The landscape of NSCLC has witnessed a paradigm shift with the recognition of a distinct biological entity characterized by exon 20 insertions within the *EGFR* gene [17]. Indeed, these variants are now considered a separate disease entity from NSCLC with *EGFR*-sensitizing mutations, posing a considerable clinical challenge as they historically fell within the realm of ‘*EGFR* positive’ tumors. The intrinsic resistance of most EGFR exon 20 insertions to conventional EGFR-TKIs is widely demonstrated, with dismal response rates (RR, 0% to 27%) and PFS (median PFS 3 months) [22,23].

Similarly, immune checkpoint inhibitors (ICIs) have not demonstrated significant efficacy in tumors with *EGFR* exon 20 insertions, with retrospective studies showing ORR 4% and PFS ranging from 2.3 to 3.1 months with ICI monotherapy [24,25,26,27].

As such, the first-line therapeutic approach for metastatic NSCLC with *EGFR* exon 20 insertions continues to rely on platinum-based chemotherapy, still with limited benefit. Real-world data reveal a nuanced scenario, with patients experiencing a median overall survival (OS) ranging from 16.2 to 24.3 months, highlighting the pressing need for tailored treatment strategies [26,28].

The use of pan-HER TKIs (namely poziotinib and tarloxotinib) and the highly selective TKI mobocertinib targeting exon 20 insertions raised interest with a median PFS reaching 7 months; however, with an ORR of 28% (Table 1) and safety concerns with high rates of toxicity (most commonly diarrhea and rash), [29,30,31,32]. Of note, after an accelerated approval was granted to mobocertinib by the Food and Drug Administration (FDA), this drug has been withdrawn from the market following the negative results of the phase 3 EXCLAIM-2 confirmatory trial in which the primary endpoint of PFS was not met [33] (Table 1).

Several molecular mechanisms related to MET acquired resistance as well as genetic aberrations or phenotypic changes could contribute to cancer progression during treatment with EGFR-TKIs, leading to the loss of EGFR expression with the inability of TKIs to work efficiently. MET pathway activation occurs following an overexpression of the MET ligand hepatocyte growth factor (HGF) and this event leads to the activation of MAPK and PI3K/AKT signaling pathways. As a result, the stimulation of oncogenic pathway signaling causes an occurrence of irreversible EGFR-TKI resistance [41,42]. To overcome resistance to targeted therapies in patients with non–small cell lung cancer, a novel approach is given by amivantamab, which belongs to the novel class of *EGFR* mesenchymal–epithelial transition factor (MET) fully human bispecific antibodies. Interestingly, due to its role in binding to the extracellular domains of *EGFR* and *MET* receptors, amivantamab is demonstrating a favorable toxicity profile against both *EGFR* exon 20 insertion tumors in pre-treated patients with NSCLC and in those harboring classical *EGFR* mutations. A targeted approach of using amivantamab is substantially able to overcome ligand-site resistance in NSCLC patients with *EGFR* exon 20 insertions and address MET-mediated resistance [43,44]. In the phase 1 CHRYSALIS trial, amivantamab elicited an ORR of 40%, a median PFS of 8.3 months, and a median OS of 22.8 months in *EGFR* exon 20 insertion-mutant NSCLC patients that were previously treated [36] and it has been granted regular approval by the FDA and EMA.

Building on the safety and efficacy findings explored in a prior study involving 20 patients with NSCLC as part of the CHRYSALIS trial [45], the phase 3 PAPILLON trial was conducted to evaluate the efficacy and safety of amivantamab in combination with carboplatin–pemetrexed in comparison to standard chemotherapy alone as a first-line treatment for patients (n = 308) with advanced NSCLC with *EGFR* exon 20 insertions [40]. The primary endpoint in this trial, PFS by independent central review, was met, with 11.4 versus 6.7 months (HR 0.40, 95% CI 0.30–0.53) in favor of the amivantamab plus chemotherapy combination across all prespecified subgroups, including those with a history of brain metastases. Of note, impressive long-term results were obtained, with 31% being progression-free at 18 months with the amivantamab combination versus only 3% in the standard chemotherapy arm, possibly due to an immune cell–directing activity of amivantamab [40]. Another impressive result is the responses. Indeed, responses with the combination occurred more frequently (ORR 73% vs. 47%; rate ratio 1.50, 95% CI 1.32–1.68), earlier (median time to response 6.7 vs. 11.4 weeks), and were more durable (9.7 vs. 4.4 months) in comparison to chemotherapy alone [40]. Regarding OS, with 33% data maturity and despite 66% of patients progressing in the chemotherapy group receiving subsequent amivantamab, a reduced risk of death was observed in the amivantamab plus chemotherapy arm (OS HR 0.67, 95% CI 0.42–1.09). Safety results in this trial were in line with that expected from each agent. The most frequent were hematologic (about 50% each for anemia and neutropenia in both arms) and paronychia and rash (56% and 54%, respectively, in the amivantamab and combination arms). Most grade 3 or higher adverse events were hematologic, associated with chemotherapy in both arms (neutropenia 33% and 23%; anemia 11% and 12%, respectively), and rash (11%) associated with amivantamab [40].

A relevant aspect remains the rate of infusion-related reactions with amivantamab (42% in the amivantamab–chemotherapy group and 1% in the chemotherapy group) [40]. However, the incidence in the PAPILLON trial was lower as compared with amivantamab monotherapy trials (67%) and remains well-manageable [36]. A subcutaneous formulation of amivantamab is under investigation in the PALOMA trials (NCT04606381, NCT05498428, and NCT05388669), which may notably reduce infusion-related reactions and infusion-related time [46].

Based on the results from the PAPILLON trial, the FDA has recently expanded the approval of amivantamab plus carboplatin/pemetrexed as a first-line treatment of patients with *EGFR* exon 20 insertion positive NSCLC, thus becoming the first approved novel treatment in this setting (https://www.cancernetwork.com/view/fda-accepts-sbla-for-amivantamab-chemo-in-egfr-advanced-metastatic-nsclc (accessed on 20 November 2023)).

Moving forward, the future will probably see the return of TKIs, with the development of novel irreversible EGFR-TKIs targeting exon 20 insertions. Three compounds have demonstrated encouraging results in early phase trials: zipalertinib (CLN-081) [47], sunvozertinib (DZD9008), and furmonertinib (AST2818). Zipalertinib demonstrated 38% ORR in a phase ½ trial, with 23% grade 3 or higher adverse events, and received breakthrough therapy designation (BTD) for use in previously treated patients with *EGFR* exon 20 insertion-positive NSCLC [39,47]. Sunvozertinib has been approved in China after the results of the WU-KONG6 trial showed ORR 61% in pretreated patients, with 45% experiencing grade 3 AEs [37]. Furmonertinib received breakthrough therapy designation (BTD) for use in previously treated patients with *EGFR* exon 20 insertion-positive NSCLC after data from the phase 1b FAVOUR trial showed ORR 69% in treatment-naïve patients, with grade ≥ 3 AEs occurring in 13–29% of patients across cohorts (Table 1). Of note, responses with these compounds were also observed in patients who previously received amivantamab, underlining their different mechanisms of action that may help overcome treatment resistance [37,38,47]. Phase 3 clinical trials are also ongoing in the first-line setting with these compounds (Table 2), and their results will shed light on the next steps forward in treating patients with *EGFR* exon 20 insertion-positive NSCLC.

## 2. Conclusions

A deep focus on molecular hallmarks has allowed us to identify and specifically distinguish a subset of advanced NSCLC patients harboring *EGFR* exon 20 insertions with the possibility of benefiting from targeted therapies, including bispecific antibodies such as amivantamab that demonstrate a high affinity and target both *EGFR* and *MET* [48]. In summary, our commentary pointed out significant clinical improvements in ORR and PFS in the PAPILLON study (amivantamab + platinum-based chemotherapy vs. chemotherapy alone), supporting the use of this regimen as the potential standard-of-care in the first-line treatment of *EGFR* exon 20 insertion-positive NSCLC patients [40]. In this context, a deeper capability to detect and categorize the diverse *EGFR* exon 20 insertion variants will be key to optimizing treatment selection and dosing in order to boost TKI efficacy and address the challenges posed by this distinct subset of NSCLC in currently ongoing and future studies.

## Figures and Tables

**Table 1 cancers-16-01331-t001:** Summary of main study results in *EGFR* exon 20 insertion-positive NSCLC.

Trial Name	Experimental Treatment	Phase	Setting	ORR	PFS	OS	G ≥ 3 AEs
ZENITH20 [34,35]	Poziotinib	2	pretreated	19.3%	4.2 m	NR	83%
EXCLAIM [31,32]	Mobocertinib	1/2	pretreated	28%	7.3 m	24 m	69%
RAIN-701 [30]	Tarloxotinib	1	pretreated	0%	NR	NR	47%
CHRYSALIS [36]	Amivantamab	1	pretreated	40%	8.3 m	22.8 m	35%
WU-KONG6 [37]	Sunvozertinib	2	pretreated	61%	NR	NR	45%
FAVOUR [38]	Furmonertinib	1b	1 L pretreated	69% 41%	10.7 m 5.8–7 m	NR	13% 18–29%
REZILIENT [39]	Zipalertinib	1/2a	pretreated	38%	10 m	NR	23%
EXCLAIM-2 NCT04129502 [33]	Mobocertinib	3	1 L	32%	9.6 m	NR	62%
PAPILLON [40]	Amivantamab + platinum-based chemotherapy	3	1 L	73%	11.4 m	NE	75%

m: months; NR: not reported; NE: not estimable.

**Table 2 cancers-16-01331-t002:** Phase 3 trials ongoing in EGFR exon 20 insertion-positive NSCLC.

Trial ID	Drug	Comparator	Setting	Primary Endpoint
NCT05668988	Sunvozertinib	Pemetrexed + carboplatin	1L	PFS
NCT05607550 FURVENT	Furmonertinib	Platinum-based chemotherapy	1L	PFS
NCT05973773 REZILIENT3	Zipalertinib + platinum-based chemotherapy	Platinum-based chemotherapy	1L	PFS

1L: first line; PFS: progression free survival.

## Data Availability

Not applicable.

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
