# Peer review of "Uncommon and Rare EGFR Mutations in Non-Small Cell Lung Cancer Patients with a Focus on Exon 20 Insertions and the Phase 3 PAPILLON Trial: The State of the Art"

_cancers, 2024, doi:10.3390/cancers16071331_

Round 1
Reviewer 1 Report
Comments and Suggestions for Authors
This is a review style comment on uncommon EGFR mutations. The content is well organized and crucial.
1.Authors should move the purpose of the comment that placed the last- “This commentary reports the real possibility to detect and treat uncommon EGFR mutations that until recently represented an unmet need, to the best of our knowledge and on the basis of the papers that we deemed important to discuss.”, to the top of the article.
2. Conclusion contenting summary is recommended at the bottom.
Author Response
We appreciate very much the time spent in reviewing our commentary entitled “Uncommon and rare EGFR mutations in non-small cell lung cancer (NSCLC) patients with a focus on exon 20 insertions and the phase 3 PAPILLON trial: the State of the Art" submitted for publication in Cancers Journal. We have read all comments’ carefully and replied point by point in the attached file.

Reviewer 2 Report
Comments and Suggestions for Authors
Although there is a noticeable effort to present data around a very interesting topic, the writing makes reading the text difficult and requires significant corrections to be considered suitable for publication
Comments on the Quality of English LanguageAlthough there is a noticeable effort to present data around a very interesting topic, the writing makes reading the text difficult and requires significant corrections to be considered suitable for publication
Author Response

(The authors gave the same response as above.)

Reviewer 3 Report
Comments and Suggestions for Authors
The commentary “Uncommon and Rare EGFR Mutations in Non-Small Cell Lung Cancer (NSCLC) Patients with A Focus on Exon 20 Insertions and the Phase 3 PAPILLON Trial: the State of the Art” by Federico Pio Fabrizio is a concise overview of the role of rare and uncommon mutations (RaUM) within a EGFR gene in NSCLC and its response to therapy. It's known that patients harboring RaUM are (most of the time) less sensitive to therapy with tyrosin kinase inhibitors (TKI) targeting aberrant EGFR. The authors provide an information on spectrum and occurrence rate of RaUM of EGFR. They overview several studies focusing on RaUM in exon 20 of EGFR and response of the carriers of such mutation to therapy with Afatinib, Poziotinib, and others. They also discuss data demonstrating better response of the carriers of some RaUM to EGFR-TKI. Particular attention of the authors in this commentary is given to PAPILLON trial (combination of amivantamab and carboplatin–pemetrexed vs standard chemotherapy as a first-line treatment) and its promising results. Next, they mention several TKI with breakthrough therapy designation (BTD) for NSCLC with RaUM in exon 20. The authors suggest that detecting a whole spectrum of EGFR exon 20 variants with the use of NGS and their categorization in terms of therapy response (including TKI therapy) will allow to optimize curative approaches.
In my opinion, the commentary is timely and will be instrumental for those working in the field.
Please find my commentaries below.
In the introduction, the information about genetics/pathology of NSCLC (and the role EGFR Mutations, in particular) is missing. The authors start the commentary from introducing NGS as a tool for genetic diagnostics, etc. I suggest adding couple of sentences about NSCLC and EGFR.
“rare and uncommon mutations” - I suggest using abbreviation
Line 22. The sentence “Their incidence is under molecular and clinical investigations according to recent findings that reported an increase of sensitivity and specificity of next-generation sequencing (NGS) methods which allow their detection and emerging treatment options that would significantly improve patients’ outcomes with these particular subgroups of EGFR-mutated advanced non-small cell lung cancer (NSCLC)” is tremendously long and its narrative is hard to follow. I encourage the authors to rewrite the sentence.
Line 127 “EGFR mesenchymal–epithelial transition factor (MET) bispecific antibody”. It might be beneficial to introduce briefly the molecular mechanisms of MET-mediated resistance to therapy.
Author Response

(The authors gave the same response as above.)

Reviewer 4 Report
Comments and Suggestions for Authors
Well written.
Accepted
Author Response

(The authors gave the same response as above.)

Round 2
Reviewer 2 Report
Comments and Suggestions for Authors
Despite the commendable effort of the authors and the interesting topic they have chosen, significant improvements in the text's syntax are necessary as it is difficult to read. The clarity is lacking in many parts of the text, and the grammatical errors are significant. Additionally, the tone seems to vary, being overly simplified at times and suitable for scientific publication at other
Comments on the Quality of English LanguageDespite the commendable effort of the authors and the interesting topic they have chosen, significant improvements in the text's syntax are necessary as it is difficult to read. The clarity is lacking in many parts of the text, and the grammatical errors are significant. Additionally, the tone seems to vary, being overly simplified at times and suitable for scientific publication at others.